# Reciprocal Regulation of TRPS1 and miR-221 in Intervertebral Disc Cells

**DOI:** 10.3390/cells8101170

**Published:** 2019-09-28

**Authors:** Letizia Penolazzi, Elisabetta Lambertini, Leticia Scussel Bergamin, Carlotta Gandini, Antonio Musio, Pasquale De Bonis, Michele Cavallo, Roberta Piva

**Affiliations:** 1Department of Biomedical and Specialty Surgical Sciences, University of Ferrara, 44121 Ferrara, Italy; maria.letizia.penolazzi@unife.it (L.P.); elisabetta.lambertini@unife.it (E.L.); scsltc@unife.it (L.S.B.); carlotta.gandini@stud.unifi.it (C.G.); 2Department of Neurosurgery, S. Anna University Hospital, 44124 Ferrara, Italy; antonio.musio1@unife.it (A.M.); pasquale.debonis@unife.it (P.D.B.); michelecavallo@hotmail.com (M.C.); 3Center for Studies on Gender Medicine, University of Ferrara, 44121 Ferrara, Italy

**Keywords:** intervertebral disc cells, intervertebral disc degeneration, TRPS1, miR-221

## Abstract

Intervertebral disc (IVD), a moderately moving joint located between the vertebrae, has a limited capacity for self-repair, and treating injured intervertebral discs remains a major challenge. The development of innovative therapies to reverse IVD degeneration relies primarily on the discovery of key molecules that, occupying critical points of regulatory mechanisms, can be proposed as potential intradiscal injectable biological agents. This study aimed to elucidate the underlying mechanism of the reciprocal regulation of two genes differently involved in IVD homeostasis, the miR-221 microRNA and the TRPS1 transcription factor. Human lumbar IVD tissue samples and IVD primary cells were used to specifically evaluate gene expression and perform functional analysis including the luciferase gene reporter assay, chromatin immunoprecipitation, cell transfection with hTRPS1 overexpression vector and antagomiR-221. A high-level expression of TRPS1 was significantly associated with a lower pathological stage, and TRPS1 overexpression strongly decreased miR-221 expression, while increasing the chondrogenic phenotype and markers of antioxidant defense and stemness. Additionally, TRPS1 was able to repress miR-221 expression by associating with its promoter and miR-221 negatively control TRPS1 expression by targeting the *TRPS1*-3′UTR gene. As a whole, these results suggest that, in IVD cells, a double-negative feedback loop between a potent chondrogenic differentiation suppressor (miR-221) and a regulator of axial skeleton development (TRPS1) exists. Our hypothesis is that the hostile degenerated IVD microenvironment may be counteracted by regenerative/reparative strategies aimed at maintaining or stimulating high levels of TRPS1 expression through inhibition of one of its negative regulators such as miR-221.

## 1. Introduction

Intervertebral disc (IVD) degeneration can occur in any area of the spine (cervical, thoracic and lumbar) for different reasons including trauma, chronic overload, aging or genetic factors. It represents a major cause of lower back pain, a leading cause of disability worldwide [1]. The IVD is a complex joint consisting of a central water-rich gelatinous tissue (rich in proteoglycans and type II collagen, the nucleus pulposus, NP), a collagen-rich fibrous lamellar structure surrounding the NP (anulus, AF), and cartilaginous endplates [2]. Degeneration is characterized by the loss of chondrocyte-like phenotype by the cells, and it is often irreversible because NP tissue has low cellularity and low regenerative capacity [1,3]. Moreover, degenerated discs have a hypoxic and inflammatory microenvironment leading to up-regulation of catabolic factors and further degeneration [4]. This microenvironment is tricky to understand from a biochemical point of view due to its great heterogeneity. Consequently, detecting potential specific therapeutic target molecules is not easy and remains a major challenge. In fact, the development of innovative effective therapies to reverse degeneration and restore mechanical and biochemical properties of the disc, relies primarily on the discovery of key molecules that occupy critical/central points of regulatory mechanisms [1,4,5]. In particular, identifying the mechanisms of transcription factor (TF) dependencies can lead to new targetable therapeutic approaches. In this study, we addressed how the TRPS1 transcription factor may function as a critical regulator in the IVD degenerated context. TRPS1, an atypical member of the GATA transcriptional factor family causing the Tricho-rhino-phalangeal syndrome type I, has been initially described as a transcriptional repressor [6]. However, it has been recently demonstrated that TRPS1 may participate in different chromatin complexes and, depending on the complex composition, may affect transcription positively or negatively [7,8].

In addition to the well documented role of TRPS1 in the onset of human cancer [8,9], important data are emerging on the involvement of this TF in the development and in maintaining tissue homeostasis, especially in bone, hair follicles and kidney [10,11]. In particular, it has been recently suggested that TRPS1 might regulate development in the axial skeleton modulating a subset of mineralization genes [12,13]; moreover, among those genes that can be used as markers to distinguish developing IVD from vertebrae in mouse TRPS1 has been included [12] as well as in the list of cartilage formation genes [14].

Relying on this evidence from the literature and in favor of the hypothesis that TRPS1 may play different roles depending on the cellular context in which it is found, we thought that this TF could be part of the recently proposed vicious circle that supports IVD degeneration through a loop involving cells that lose their chondrocyte-like phenotype, extracellular matrix, metabolites and biomechanics [15]. The purpose of this study was to understand through which mechanisms TRPS1 could act in IVD cells, since the final aim falls within the project of developing regenerative/reparative strategies for the treatment of IVD degeneration through detection of good candidates as potential intradiscal injectable molecule [5,16]. Previously, we have observed a dramatic decrease of degenerated phenotype of IVD cells after silencing of a potent antichondrogenic microRNA, miR-221 [17], accompanied by the increase of TRPS1 expression [17,18]. We are interested here in deepening this aspect and understanding if a regulatory axis TRPS1-miR-221 may be critical in the maintenance of disc cell functions. It is important to underline that microRNAs represent an emerging area of research that will provide new insight into IVD degeneration since some of these post-transcriptional gene regulatory elements have been shown to be involved in multiple pathological processes during disc degeneration, including apoptosis, ECM degradation, cell proliferation and inflammatory response [19,20].

In this study we investigated possible mechanisms underlying the inverse regulation between miR-221 and TRPS1, also exploring potential molecules that lie upstream and downstream of TRPS1 signaling, and that are involved on the ability of lumbar IVD cells to differentiate, maintain the antioxidant defense and stemness.

## 2. Materials and Methods

### 2.1. Patients and Tissue Samples

Human lumbar disc tissues were obtained from 30 donors (patients’ age was between 37 and 79 years, mean age 56 years, 18 males and 12 females, see Table 1) by using research protocol approved by Ethics Committee of the University of Ferrara and S. Anna Hospital (protocol approved on 17 November, 2016). Patients were operated for the herniated lumbar disc through a microsurgical posterior approach. Disc sampling was obtained from the central core of the disc, in order to avoid anterior and posterior longitudinal ligament, anulus and calcified portions of the disc.

The level of disc degeneration was evaluated analyzing the signal characteristics of the disc in T2-weighted MRI (magnetic resonance imaging) sequences according to Pfirrmann classification [21].

In particular, this grading system consists of five grades of lumbar disc degeneration. Four different parameters were analyzed: Homogeneity of the disc, height of the disc, intensity of disc’s signal on the MRI and distinction between the nucleus and anulus. From I grade to V grade of this classification, the lumbar disc appears from homogeneous to inhomogeneous respectively, it decreases its height, from a bright hyperintense white signal intensity it gains a hypointense black signal intensity and the distinction between the nucleus and anulus is progressively lost.

Every lumbar vertebral disc sample was immediately preserved in sterile saline solution and processed within 24 h from the surgery.

### 2.2. Isolation of Human IVD Cells

Lumbar intervertebral disc tissues (1–2 cm^3^) were collected, cut into small pieces and subjected to mild digestion in 15 mL centrifuge tube with only 1 mg/mL type IV collagenase (Sigma Aldrich, St. Louis, MO, USA) for 5 h at 37 °C in Dulbecco’s modified Eagle’s medium (DMEM)/F12 (Euroclone S.p.A., Milan, Italy) as previously described [18]. Once the digestion was terminated, cell suspension was filtered with a Falcon™ 70 μm Nylon Cell strainer (BD Biosciences, Franklin Lakes, NJ, USA). Subsequently 300× *g* centrifugation was conducted for 10 min, the supernatant discarded, the cells resuspended in basal medium (DMEM/F12 containing 10% fetal calf serum (FCS), 100 mg/mL streptomycin, 100 U/mL penicillin and 1% glutamine; Euroclone) and seeded in polystyrene culture plates (Sarstedt, Nümbrecht, Germany) at 10000 cells/cm^2^. The cells that were released from the dissected tissue and maintained in culture at 37 °C in a humidified atmosphere with 5% CO_2_ within the first 48 h were referred to as passage zero (P0) cells. P0 cells were expanded by growing for a period not exceeding a week until subconfluent, detaching by trypsinization and maintained in culture for two passages to obtain P2 cells that were used for later experiments.

### 2.3. Cell Transfection

IVD cells were seeded in polystyrene culture plates (1.82 cm^2^ area; Sarstedt) until reaching 70% of confluence. For hTRPS1 overexpression, cells were transfected with 0.4 μg/cm^2^ of pBlight-FLAG-TRPS1 expression vector or with the empty vector (Genentech, San Francisco, CA USA) for 48 h. Then, total RNA was extracted and stored at −80 °C for subsequent quantitative RT-PCR analysis. For immunocytochemistry analysis, cells were fixed with methanol and analyzed as indicated below.

### 2.4. Luciferase Reporter Gene Assay

The 739 bp fragment of the human *TRPS1* 3′-UTR (+1233 to +1972) containing the high conserved 8-mer site for miR-221-3p (+1630 to +1638, 5′-AUGUAGCA-3′), was inserted into the XhoI-XbaI restriction sites in the multiple cloning site of the reporter vector pmiRNano-GLO (Promega Corp., Fitchburg, WA, USA). This bicistronic vector contains NanoLuc luciferase (NlucP) as the primary reporter gene and Firefly luciferase (Luc2) as control reporter for normalization. Primers used in the PCR were: Forward *TRPS1* 3′-UTR: TCTCGAGGCTCAGGGAAATAGGGCTAAA, which contains the XhoI restriction site (CTCGAG), Reverse *TRPS1* 3′-UTR GCTCTAGAGCAGATTCCAGCAACACTTATC, which contains the XbaI restriction site (TCTAGA). IVD cells were transfected with 100 ng of reporter vector in combination with 30 nM of anti-miR-221 (GAAACCCAGCAGACAAUGUAGCU), or negative control (all purchased from Ambion Life Technologies, Grand Island, WA, USA), using Lipofectamine 2000 reagent (ThermoFisher Scientific, WA, USA). After 48 h, transfected IVD cells underwent NanoLuc and Firefly luciferase activity measurements using the GloMax 20/20 single tube Luminometer (Promega Corp, Fitchburg, WA, USA) and the Nano-Glo Dual-Luciferase Assay (Promega Corp, Fitchburg, WA, USA) according to the manufacturer’s recommendations. The ratio NanoLuc reporter activity/Firefly control reporter activity was calculated for each well. For each IVD samples (*n* = 6) all transfections were performed in triplicate, and data were presented as mean values with standard deviation.

### 2.5. Immunocytochemistry

Immunocytochemistry analysis was performed employing the ImmPRESS (#MP-7500; Vectorlabs, Burlingame, CA, USA). Cells grown in polystyrene culture plates were fixed in cold 100% methanol and permeabilized with 0.2% (*v*/*v*) Triton X-100 (Sigma Aldrich, St. Louis, MO, USA) in TBS (Tris-buffered saline). Cells were treated with 3% H_2_O_2_ in TBS, and incubated in 2% normal horse serum (Vectorlabs) for 15 min at room temperature. After the incubation in blocking serum, the different primary antibodies were added and incubated at 4 °C overnight: Polyclonal antibodies for COL2A1 (#Ab3092; mouse anti-human, 1:200 dilution, Abcam, Cambridge, UK), SOX9 (#sc-20095; rabbit anti-human, 1:500 dilution, Santa Cruz Biotechnology, Dallas, TX, USA) ACAN (#sc-33695; mouse anti-human, 1:200 dilution, Santa Cruz Biotechnology), TRPS1 (#20003-1-AP; rabbit anti-human, 1:100 dilution; Proteintech Group, Rosemont, WA, USA), COL1 (#sc-28657; rabbit anti-human, 1:200 dilution, Santa Cruz Biotechnology), SOX-2 (#sc-365823; mouse anti-human, 1:500 dilution, Santa Cruz Biotechnology) and SOD2 (#sc-133134; mouse anti-human; 1:200 dilution; Santa Cruz Biotechnology). Cells were then incubated in Vecstain ABC (#MP-7500; Vectorlabs) with DAB solution (#SK-4105; Vectorlabs). After washing, cells were mounted in glycerol/TBS 9:1 and observed using a Leitz microscope (Wetzlar, Germany). Quantitative image analysis of immunostained cells was obtained by a computerized video-camera-based image-analysis system (with NIH USA ImageJ software, public domain available at: http://rsb.info.nih.gov/nih-image/) under brightfield microscopy. Briefly, images were grabbed with single stain, without carrying out nuclear counterstaining with hematoxylin and unaltered TIFF images were digitized and converted to black and white picture to evaluate the distribution of relative gray values (i.e., number of pixels in the image as a function of gray value 0–256), which reflected chromogen stain intensity. Images were then segmented using a consistent arbitrary threshold of 50% to avoid a floor or ceiling effect, and binarized (black versus white); total black pixels per field were counted and average values were calculated for each sample. Three replicate samples and at least ten fields per replicate were subjected to densitometric analysis. We performed the quantification of pixels per 100 cells and not per area in order to take into account the different cell morphology and confluence.

### 2.6. RNA Extraction and Quantitative Real-Time (qRT)-PCR

Total RNA, including miRNAs, was extracted from IVD cells using the RNeasy Micro Kit (Qiagen, Hilden, Germany), according to the manufacturer’s instructions. RNA concentration and quality were measured using a NanoDrop ND1000 UV-VIS spectrophotometer (Isogen Life Science, de Meern, the Netherlands). cDNA was synthesized from total RNA in a 20 μL reaction volume using the TaqMan MicroRNA Reverse Transcription kit (ThermoFisher Scientific, WA, USA) for analysis of microRNA. Quantification of miR-221 was performed using the TaqMan MicroRNA Assays (ThermoFisher), using U6 snRNA for normalization. Polymerase chain reactions were performed with the TaqMan Universal PCR MasterMix (ThermoFisher) using the CFX96TM PCR detection system (Bio-Rad, Hercules, CA, USA). Relative gene expression was calculated using the comparative 2^−△Ct^ or 2^−△△Ct^ method, where indicated. All reactions were performed in triplicate and the experiments were repeated at least six times.

### 2.7. Histochemical Analysis

Small fragments of each IVD sample were rinsed with PBS, fixed in 4% buffered paraformaldehyde for 24 h at 4 °C, embedded in paraffin and cross-sectioned (5 μm thick). For histological evaluation non-consecutive sections were immunostained with TRPS1 (#20003-1-AP; rabbit anti-human, 1:100 dilution; Proteintech, Rosemont, IL, USA) and SOD-2 (#sc-133134; mouse anti-human; 1:100 dilution; Santa Cruz Biotechnology). Immunohistochemical sections were deparaffinized, rehydrated and heated in sodium citrate (pH 6) for antigen retrieval. Slides were then processed with 3% H_2_O_2_ in PBS for 5 min and with blocking solution (PBS /1% BSA/10% FCS) for 30 min at room temperature. Then the slides were incubated over night with the primary antibody at 4 °C, followed by treatment with Vectastain ABC solution (Vectorlabs) for 30 min. The reactions were developed using DAB solution (Vectorlabs), the sections were counterstained with hematoxylin and mounted in glycerol. The stainings were quantified by a computerized video camera-based image analysis system (NIH, USA ImageJ software, public domain available at: http://rsb.info.nih.gov/nih-image/) under brightfield microscopy (NikonEclipse 50i; Nikon Corporation, Tokyo, Japan), as reported above. For the analysis of sections, positive cells in the area were counted and protein levels expressed as % of positive nuclei (ten fields per replicate, five sections per sample).

### 2.8. In Silico Analysis of the miR221/222 Human Promoter

The miR-221/222 cluster is intergenic and positioned in chromosome X. The prediction of TRPS1 binding sites in the human promoter region of miR221/222 (from +1192 to −3500 bp) was performed by using Alibaba2.1 (http://gene-regulation.com/pub/programs/alibaba2/index.html) and PROMO (http://alggen.lsi.upc.es/cgi-bin/promo_v3/promo/promoinit.cgi?dirDB=TF_8.3) public software. We observed nine conserved TRPS1 binding sites in its immediate upstream promoter and four downstream of the +1 site (Figure 3).

### 2.9. Chromatin Immunoprecipitation (ChIP) Assay

Chromatin immunoprecipitation (ChIP) was performed by using Magna ChIP Protein A+G magnetic beads (Merck Millipore, Billerica, MA, USA) according to the manufacturer’s instructions and as previously described [22,23]. Briefly, IVD progenitors were cross-linked with 1% formaldehyde at 37 °C for 10 min. The cells were washed in ice-cold PBS, and resuspended in SDS lysis buffer supplemented with 1X protease inhibitor cocktail (Sigma Aldrich) for 10 min on ice. The isolated chromatin was sonicated to an average size of about 200–1000 bp. Immunoprecipitation reactions were performed by incubating the chromatin with Protein A+G magnetic beads, 5 mg of antibody against TRPS1 (#20003-1-AP; rabbit anti-human; Proteintech) or control IgG at 4 °C for 16 h. Immunoprecipitated chromatin complexes were sequentially washed with 1 mL each of the following buffers: Low salt wash buffer (0.1% SDS, 1% Triton X-100, 2 mM EDTA, 20 mM Tris–HCl pH 8.1 and 150 mM NaCl), high salt wash buffer (0.1% SDS, 1% Triton X-100, 2 mM EDTA, 20 mM Tris–HCl pH 8.1 and 500 mM NaCl), LiCl wash buffer (0.25 mM LiCl, 1% IGEPAL-CA630, 1% deoxycholic acid, 1 mM EDTA and 10 mM Tris pH 8.1) and TE buffer (10 mM Tris-HCl pH 7.4 and 1 mM EDTA). The immunocomplexes were eluted by adding a 200 μL aliquot of a freshly prepared solution of 1% SDS, 0.1 M NaHCO_3_ followed by incubation at 65 °C for 30 min. The crosslinking reactions were reversed by addition of 0.2 M NaCl (final concentration) followed by incubation at 65 °C for 16 h. The samples were then digested with proteinase K (10 mg/mL) at 42 °C for 1 h and DNA was purified in 50 μL of Tris–EDTA with a PCR purification kit (Promega) according to the manufacturer’s instructions. For PCR analysis, aliquots of chromatin before immunoprecipitation were saved (Input). PCR was performed by using specific primers (listed in Table 2) to amplify different regions of the miR-221 promoter (see Figure 3).

Real-time PCR analyses of the ChIP samples were carried with CFX96 Real-Time detection System (Bio-Rad labs) using iTaqUniversal SYBR Green SuperMix (Bio-Rad, Hercules, CA, USA). We analyzed ChIP-qPCR data relative to the input signal and presented as a fold increase in the signal relative to the background signal (IgG).

### 2.10. Statistical Analysis

Data are reported as mean value ± standard deviation (SD). Unpaired Student’s *t*-test was used for direct comparisons; multiple groups were compared by using one-way ANOVA, followed by Tukey’s HSD post-hoc test. All statistical analyses were performed using Prism 6 software (GraphPad Software, Inc., Version 5.0, San Diego, CA, USA). *p*-values less than 0.05 were considered significant.

## 3. Results

### 3.1. The Expression of TRPS1 in Lumbar Degenerated IVD Tissues

We collected IVD tissue samples from 30 cases (Table 1), and measured the expression of TRPS1 in each specimen. TRPS1 expression was found in all samples regardless of age and sex of the patients, predominantly in the nucleus. As shown in Figure 1A, TRPS1 expression levels significantly decreased with increasing grade of disc degeneration. Highly degenerated discs, classified according to Pfirrmann grading system, expressed low levels of TRPS1, on the contrary a high-level expression of TRPS1 was significantly associated with the lower pathological stage. It is noteworthy that we did not observe this association in the IVD from the cervical spine, which expressed TRPS1 at comparable levels regardless of the degree of degeneration [18].

As shown in Figure 1B, the reduction of TRPS1 was concomitant with a decrease in the expression of manganese-containing superoxide dismutase (SOD2), an enzyme with an important role in cellular stress responses and implicated in antiapoptotic action and defense against oxidative stress [24]. SOD2 has been recently demonstrated to be an important effector of FOXO signaling in disc degeneration, and its reduction is correlated with decrease in the expression of FOXO transcription factors [25]. Therefore, the TRPS1 decrease observed by us could be considered part of the biomolecular damage accumulation.

### 3.2. TRPS1 Signaling and miR-221

In order to investigate the role of TRPS1-mediated signaling, IVD primary cell cultures were set up from samples with different Pfirrmann grades and essentially consisting of nucleus pulposus, as reported in the Material and Methods section. In order to obtain the most informative results from the endogenous degenerated microenvironment, we chose to preserve the whole cell population deriving from the biopsy without performing cell sorting.

As argued in a previous paper [18], we used passage two (P2) cells since they represent a good compromise as de-differentiated but no senescent cells. In these cells we assessed the effects of TRPS1 overexpression in terms of differentiation, oxidative stress and antioxidant defense. Although disc cell phenotypes still remain to be defined in detail, it is accepted that IVD cells exhibit a chondrocyte-like phenotype. As shown in Figure 2A, overexpression of TRPS1 significantly increased the expression of chondrogenic markers such as sex determining region Y box 9 (SOX9), type II collagen (COL2A1) and aggrecan (ACAN), demonstrating the positive role of this transcription factor in restoring the chondrocyte-like phenotype that had been lost during the de-differentiation process. The expression level of these chondrogenic markers in TRPS1 overexpressing IVD cells were comparable with those by us detected in human primary chondrocytes from nasal septum (data not shown) [17]. Consistently, TRPS1 overexpression had no effect on type I collagen (COL1) expression (Figure 2A).

This was strengthened by the demonstration that the expression of a potent antichondrogenic factor such as miR-221 [26] was almost completely abolished after TRPS1 overexpression (Figure 2B).

The expression of SOD2 was also found increased in TRPS1 overexpressing cells (Figure 2C), further confirming the immunohistochemical data (Figure 1B) and the positive effect of TRPS1 in IVD cells.

Notably, as shown in Figure 2C, TRPS1 overexpressing cells significantly increased the expression level of SOX-2, which is an essential transcription factor for self-renewal and survival of progenitor/stem cells [27], and may support the action of resident progenitors present in the stem cell niche recently found in the IVD [28].

In order to assign to TRPS1 a key role in the pro-chondrogenic pathway and to gain further insight into the mechanism of suppression by it mediated, we investigated if miR-221 decreased expression in TRPS1 overexpressing cells could be mediated by TRPS1 binding to the miR-221 promoter. Indeed, searching of the miR-221 promoter region for transcription factor binding sites by using Alibaba2.1 and PROMO programs, nine conserved consensus potential TRPS1 binding sites in its immediate upstream promoter and four downstream of the +1 site were identified (Figure 3). To test TRPS1 recruitment to this region, a preliminary chromatin immunoprecipitation (ChIP) analysis was performed in four samples. By using the specific primers reported in Figure 3, four different regulatory regions containing the potential TRPS1 binding sites were examined for their ability to recruit TRPS1.

Each sample was separately evaluated. The results revealed that (i) TRPS1 recruitment occurred in three out of four patients, (ii) the promoter region 4 resulted in not being accessible, (iii) the other regulatory regions were differently involved in TRPS1 binding and (iiii) the absence of TRPS1 binding or the recruitment at only one region was found in the samples with a higher level of degeneration.

A thorough epigenetic investigation is necessary to show that regulation of miR-221 expression by TRPS1 is significant in IVD cells, however overexpression data and ChIP analysis collectively support the hypothesis of a TRPS1-miR-221 axis, and strengthened our previous evidences on direct correlation between the downregulation of miR-221 and cartilage formation [25], and the efficacy of antagomiR-221 cell treatment in increasing TRPS1 expression [18].

### 3.3. miR-221 Directly Targets TRPS1

Unlike what was by us observed in freshly isolated (passage zero, P0) cells from cervical IVD samples [18], cells from lumbar IVD expressed substantial miR-221 at comparable levels regardless of the grade of degeneration (Figure 4A). Having, however, shown that miR-221 is highly sensitive to TRPS1 levels (Figure 2), we investigated if TRPS1 is a direct target of miR-221. We searched for potential miR-221 binding sites in the TRPS1 3′-UTR, by using three different miRNA databases (mirTarBase, DIANA-microTv5.0 and microRNA.org). As shown in Figure 4B, computational analysis identified only one conserved seed sequence for miR-221, previously validated in breast cancer cells by Stinson et al. [29]. 3′-UTR DNA fragment containing the potential miR-221 target recognition site was then generated by PCR, and cloned into the 3′-UTR of a luciferase reporter gene. The construct was then transfected in IVD cells together with a specific antagomiR oligonucleotide (anti-miR-221). As shown in Figure 4C, miR-221 depletion significantly increased the reporter gene activity, suggesting that miR-221 negatively regulates TRPS1 expression. AntagomiR-221 was highly effective in silencing miR-221, since about 99% decrease of miR-221 was detected by RT-qPCR in antagomiR-221 treated cells compared with control cells (Figure 4D).

## 4. Discussion

Despite intense investigation, the pathophysiology of IVD degeneration, which is triggered by ageing, mechanical stress, traumatic injury, infection and inflammation is still not clear [1,2]. It is widely accepted that the extracellular matrix of the degenerated disc undergoes significant modifications of proteoglycan composition and structure, as a consequence of loss and phenotypic changes of disc cells [1,2]. Likewise, it is known that the degenerated IVD microenvironment is characterized by increased expression of a pro-inflammatory/catabolic cytokines and inflammatory mediators, including IL-1, IL-6, IL-12 IL-17, TNF-α (alpha) and IFN-γ (gamma) [30]. However, little is known about the molecular regulators that drive cellular changes, and whose activity can shift the balance between self-repair capacity and microenvironment deterioration. The knowledge of specific key regulators that support the degenerative process from a molecular perspective should allow for not only better phenotyping of cells, but also developing new regenerative/reparative cell-based or biological therapies for this debilitating condition [3,4,5,31]. In recent years, research on gene expression modulation during both physiological and disease processes of IVD is focusing on attributing critical roles to transcription factors and non-coding RNAs (microRNAs and long non-coding RNAs), not only for what concerns their impact in diverse target genes, but also their interplay [32,33]. We focused here on this aspect and reported, for the first time, that the reciprocal regulation between the transcription factor TRPS1 and miR-221 seems to be of crucial importance for the maintenance of the IVD homeostasis and disc cell functions. This study originated from two key observations: 1. highly degenerated discs expressed low levels of TRPS1, on the contrary high-level expression of TRPS1 was significantly associated with the lower pathological stage and 2. IVD cells benefit from TRPS1 over-expression that accompanies the loss of the antichondrogenic miR-221 [17,26]. The validation of the relationship between these two molecules was the results of a Luc assay and preliminary ChIP experiments on human primary IVD cells, demonstrating that miR-221 directly targets 3′-UTR of the *TRPS1* gene and TRPS1 is in vivo recruited at the miR-221 promoter region. It is conceivable that these two molecules belong to a group of not yet well-identified molecular regulators whose dysfunction can cause and propagate IVD degeneration. This hypothesis is strengthened by the fact that the chondroprotective effect of TRPS1 over-expression is associated with expression increase of i. standard chondrogenic markers such as COL2A1, SOX9 and ACAN that are also required for discogenic differentiation [34], and ii. two important molecules in the tissue regeneration and repair, SOD2, which is crucial to defend cell against oxidative stress [24,25], and SOX2, a documented stemness regulator [27]. It is important to underline that SOD2 is the major mitochondrial antioxidant protein and plays a key role in regulating oxidative stress resistance [35]. Therefore, an increase in the expression of this protein can support the ability of IVD cells to neutralize reactive oxygen species, and decrease oxidative damage. Regarding SOX2, the results we obtained are preliminary and have to be confirmed by analyzing other stemness markers. However, our data suggest that, in the context here analyzed, a high level of TRPS1 could be beneficial to support the action of the resident stem cell population, increase the IVD regenerative potential and consequently positively impact IVD tissue engineering strategies. Although research into the stem cells/progenitors in the IVD is still in its infancy, recent studies have in fact shown the presence of a potential stem cell niche in the IVD [28].

It will be particularly interesting to understand how TRPS1 may impact on the discogenic phenotype. However, this is a complicated issue since the transcriptional machinery underpinning discogenic differentiation remains relatively undefined. In fact, there is still an open debate in defining specific markers of discogenic differentiation and in understanding what the original discogenic cell phenotype is [34,36,37,38].

TRPS1 is known to be involved in a delicate balance between the activities of several molecules and that different types of contextual determinants shape the TRPS1-mediated transcriptional response in a cell [7,8,9,10,11,12,13,14]. In the skeletal context TRPS1 is required for the modulation of expression of multiple genes that support bone development, cartilage formation, chondrogenesis as well as osteogenesis and the mineralization process [11,13,14,39]. It has been recently identified among those genes that can be used as markers to distinguish developing IVD from vertebrae in mouse [12]. However, the participation of TRPS1 in the molecular mechanisms that govern human development of the IVD, IVD homeostasis and function remains to be understood. In this scenario, the results we obtained may provide novel insight into how the loss of TRPS1 expression contributes to IVD degeneration, and how the positive action of TRPS1 can also be carried out by turning off miR-221. It is well known that miR-221 is a paralog of miR-222 [40]. These two miRNAs are encoded by a gene cluster, have the same seed sequence and share common predicted target genes [26,40]. miR-222 has been found upregulated in human degenerative NP cells [20] and intervertebral disc degenerated tissue [41] being closely related to the process of IVD degeneration [42,43]. Therefore, it is reasonable to expect that in our experimental model miR-222 will be subjected to the same regulatory mechanisms as miR-221. However, this deserves to be analyzed in details.

Data of the current study extend our previous finding that inhibiting miR-221 expression increased TRPS1 expression and restored the expression of FOXO3 that has been recently defined as critical mediator of IVD integrity and function, attenuating the severity grade of IVD degeneration [18]. This opens the way to explore in more detail the participation of specific gene transcription regulators to a vicious circle that supports the degeneration of the disc. The reference concerns the degenerative circle of intervertebral disc degeneration that has been recently depicted relying on interaction between cells, extracellular matrix, and biomechanics [15], with a particular emphasis to alteration/damage of extracellular matrix components. To determine the causative role of expression levels of TRPS1, miR-221 and FOXO3 in driving disc degeneration or homeostasis, and to provide more prospects for therapeutic targets, in vivo strategies will be needed. Further studies are needed also to investigate possible relationships of these molecules with perturbations of Wnt/β-catenin, Notch, interferon-alpha, hypoxia, nuclear factor kappa B (NF-kB), Sirtuin 3 and mitogen-activated protein kinase (MAPK) pathways that have been identified as important regulators of progress of disc degeneration [7,44,45,46,47,48,49].

A last aspect that will be worthy of further investigation regards the possibility that aberrant gene regulation occurs differently in diseased degenerative discs and normal aged discs, and in different regions of the degenerated spine [50,51,52,53,54,55]. This hypothesis is supported by the evidence we found examining cervical and lumbar IVDs. TRPS1 was expressed by cervical discs at comparable levels regardless of the degree of degeneration [18], whereas highly degenerated lumbar discs (Pfirrmann grades IV–V) expressed low levels of TRPS1 in respect to discs with a lower pathological stage. On the contrary, cells from the lumbar disc expressed miR-221 at comparable levels regardless of the degree of degeneration, whereas high expression levels of miR-221 were found in highly degenerated IVDs from the cervical spine. Although this observation deserves to be studied in depth, however the hypothesis that IVD from different spine regions may have molecular specific characteristics is to be kept in mind, especially in relation to the use of these data for the development of targeted therapies for diseases affecting neck and low back [3,4,5,56].

In conclusion, our results provide us with a new target for the treatment of disc degeneration suggesting that the hostile degenerated IVD microenvironment may be counteracted by regenerative/reparative strategies aimed at maintaining or stimulating high levels of TRPS1 through inhibition of one of its negative regulators such as miR-221.

## Figures and Tables

**Figure 1 cells-08-01170-f001:**
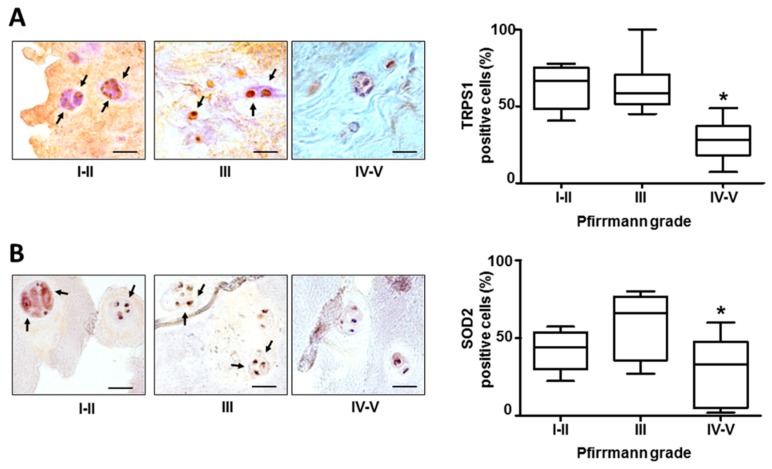
TRPS1 and SOD2 expression in IVD tissues. Immunohistochemistry of TRPS1 (**A**) and SOD2 (**B**) was performed on IVD tissues with different Pfirrmann grades of degeneration. Positive cells in representative optical photomicrographs are indicated with arrows. Protein levels were quantified by densitometric analysis of immunostaining using ImageJ software and expressed as % of positive cells per area (five sections per sample; Pfirrmann I–II group, *n* = 8; Pfirrmann III group, *n* = 9 and Pfirrmann IV–V group, *n* = 13). The results are reported as a whisker box plot representing the min to max (line indicates median). * = *p* < 0.01 (Pfirrmann IV–V group vs. Pfirrmann I–II group and Pfirrmann III group). Scale bars: 20 µm.

**Figure 2 cells-08-01170-f002:**
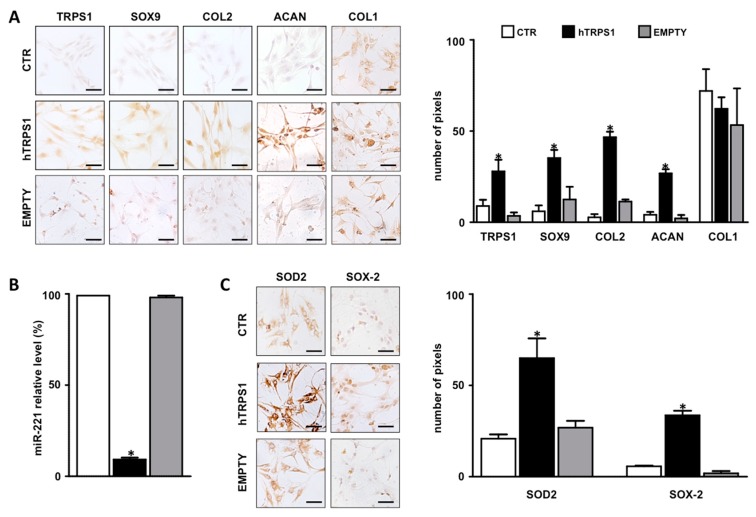
Evaluation of the effect of TRPS1 overexpression on IVD cells. Cells were transfected with the vector containing the full-length human TRPS1 cDNA (black column), with the empty vector (gray column) or remained untreated (CTR, control, white column) and harvested after 48 h. The expression of differentiation markers (**A**), miR-221 (**B**), antioxidant defense and stemness markers (**C**) were evaluated. Representative optical photomicrographs of TRPS1, COL2, SOX9, ACAN, COL1, SOD2 and SOX-2 respectively immunostaining is reported. Protein levels were quantified by densitometric analysis of immunocytochemical pictures using ImageJ software and expressed as means of pixels per one hundred cells ± SD (*n* = 10). * = *p* < 0.01. Scale bars: 20 µm. The expression levels of miR-221 were measured by qRT-PCR by the 2^(−△△Ct)^ method. Data are presented as fold changes relative to control untreated cells ± SD (*n* = 10). * = *p* < 0.01 (hTRPS1 treatment vs. CTR and empty).

**Figure 3 cells-08-01170-f003:**
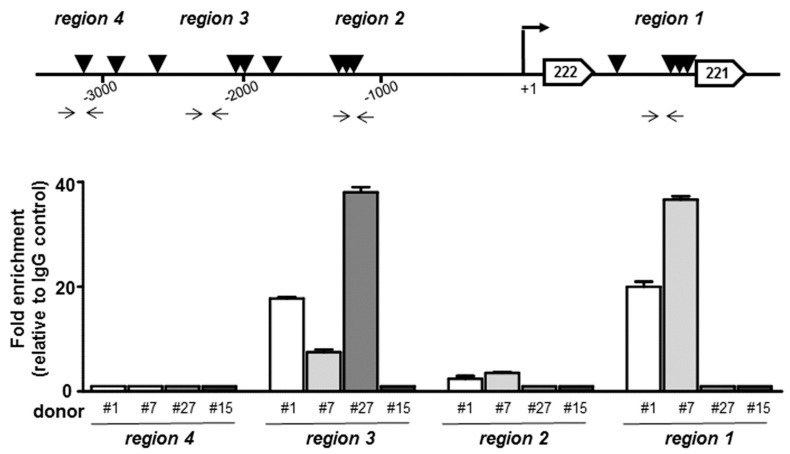
TRPS1 is in vivo recruited at the miR-221 promoter region in IVD cells. The positioning of conserved consensus potential TRPS1 binding sites within the miR-221 regulatory regions together with the position of the specific primers used for qPCR amplifications of anti-TRPS1 immunoprecipitated chromatin are reported. The graph shows the results of ChIP-qPCR analysis performed in triplicate on DNA templates obtained from four samples (donors 1, 7, 27 and 15) in relation to the four indicated regions. Results of qPCR were analyzed by the 2^(−△△Ct)^ method, normalized for the input signal and presented as a fold increase (mean ± SD) relative to the background signal (IgG).

**Figure 4 cells-08-01170-f004:**
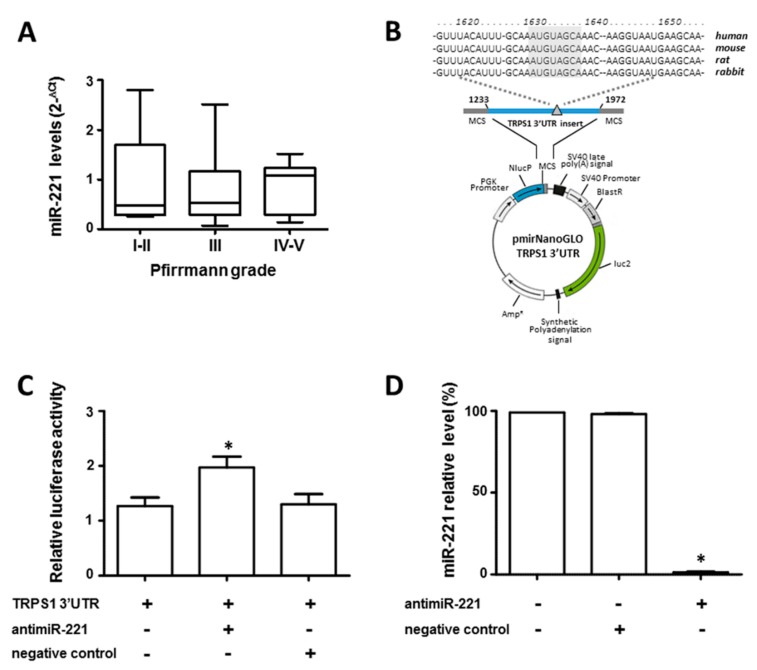
miR-221 directly targets TRPS1. The endogenous expression level of miR-221 was evaluated by qRT-PCR in the cells from IVD with a different Pfirrmann grade of degeneration (**A**). The relative miR-221 expression levels were reported by using 2^−△Ct^ method (U6 snRNA was employed for normalization; all reactions were performed in triplicate, *n* = 30 (Pfirrmann I-II group, *n* = 8; Pfirrmann III group, *n* = 9 and Pfirrmann IV-V group, *n* = 13). Validation of miR-221 target site in the TRPS1 3′-UTR by reporter gene assay in IVD cells (**B**). Schematic representation of the luciferase construct utilized in this study is reported; partial sequences of the TRPS1 mRNA 3′-UTR harboring the predicted miRNA target site (gray triangle) was inserted in the indicated position. In the upper part of the panel the position of the predicted highly conserved miRNA target site is indicated. IVD cells were transfected for 48 h with a combination of reporter constructs (100 ng) along with antimiR-221 or negative control (**C**). Afterwards, the Nano Luc luciferase reporter gene (*NlucP*) and Firefly luciferase control reporter activities (luc2) were measured using a Nano-Glo Dual-Luciferase assay and represented as mean ± SD (*n* = 6). The efficiency of miR-221 silencing after antimiR-221 treatment is reported (**D**). The expression levels of miR-221 were measured by qRT-PCR. Data are presented as fold changes relative to untreated cells ± SD (*n* = 6), * = *p* < 0.01.

**Table 1 cells-08-01170-t001:** Human intervertebral disc (IVD) samples information.

Donor	IVD Level	Age	Sex	Symptoms	Duration of Symptoms Prior to Surgery	Pfirrmann Grade
**#1**	L4L5	51	female	radiculopathy: pain and palsy, neurogenic claudication	5 months	III
**#2**	L5S1	57	male	radiculopathy: palsy	5 months	IV
**#3**	L2L3	56	male	radiculopathy: pain and palsy	2 months	II
**#4**	L4L5	49	female	radiculopathy: pain	2 months	II
**#5**	L4L5	52	male	radiculopathy: pain and palsy	1 month	III
**#6**	L4L5	79	male	radiculopathy: pain and palsy	5 months	IV
**#7**	L4L5	47	female	radiculopathy: pain	7 months	II
**#8**	L4L5	54	female	back pain and claudicatio	5 months	IV
**#9**	L5S1	44	male	radiculopathy: pain	2 months	II
**#10**	L5S1	63	female	back pain and radiculopathy: pain and palsy	12 months	V
**#11**	L4L5	57	female	claudicatio	24 months	V
**#12**	L4L5	70	male	radiculopathy: pain and palsy	1 month	IV
**#13**	L4L5	40	female	radiculopathy: pain	5 months	II
**#14**	L5S1	38	male	back pain and radiculopathy: pain and palsy	12 months	IV
**#15**	L4L5	74	female	radiculopathy: pain and neurogenic claudication	12 months	V
**#16**	L4L5	70	male	radiculopathy: pain and palsy	2 months	IV
**#17**	L3L4	56	male	radiculopathy: pain	3 months	I
**#18**	L4L5	46	male	radiculopathy: pain and palsy	12 months	III
**#19**	L5S1	47	female	radiculopathy: pain	2 months	IV
**#20**	L5S1	51	male	radiculopathy: pain	6 months	IV
**#21**	L4L5	51	male	radiculopathy: pain	3 months	III
**#22**	L5S1	63	female	radiculopathy: pain and palsy	1 month	III
**#23**	L4L5	37	male	radiculopathy: pain and palsy	2 months	II
**#24**	L5S1	49	male	radiculopathy: pain and palsy	9 months	III
**#25**	L5S1	44	male	radiculopathy: pain	4 months	III
**#26**	L4L5	54	male	radiculopathy: pain	2 months	III
**#27**	L4L5	64	male	radiculopathy: pain and palsy	3 months	IV
**#28**	L4L5	56	female	radiculopathy: pain and neurogenic claudication	3 months	IV
**#29**	L4L5	77	male	radiculopathy: pain and palsy, neurogenic claudication	10 months	III
**#30**	L4L5	72	female	radiculopathy: pain and palsy	12 months	II

**Table 2 cells-08-01170-t002:** PCR primers used for chromatin immunoprecipitation assay (ChIP).

Region	Location	Primer Sequences
Region 1	+572/+727	F-GGTATCATTTGGATAGATCAAT R-TGGATGGAAGGAAGGTCGGATAGATAA
Region 2	−1217/−1058	F-AGCCACTTTTCTCTTGGTGAT R-CGTCTTAGAATCCTTTGCTGTG
Region 3	−2419/−2234	F-CCTGTTCTAACCGTGTGGAGT R-CCATACATTCTGGCTAAAGACC
Region 4	−3429/−3140	F-GGAATCCAAGTTCATAAGAACA R-ATGGTGATGGTATCACAGGTG

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
