# Peer review of "Reciprocal Regulation of TRPS1 and miR-221 in Intervertebral Disc Cells"

_cells, 2019, doi:10.3390/cells8101170_

Round 1
Reviewer 1 Report
This is an original article by Penolazzi and colleagues that seeks to further characterize the role of TRPS1 and miR-221 in IVD homeostasis and degeneration. The study is based on previous published data by the group showing that miR-221 promotes IVD degeneration in part through suppression of TRPS1 and FOXO3. However, the data provided in the present follow up study is not entirely novel and just incremental over what the group has already published. Most of the claims in this manuscript are not sufficiently supported by data.
Below are my major concerns with the manuscript:
- Sample collection is a major concern. It is not clearly indicated whether the isolated cells are just NP or a mix of NP+AF cells. This is a major limitation that can greatly confound the results.
- Correlation analysis of TRPS1 expression with different sample parameters (grade, age, region of the spine) is indicated in the text but no data is shown.
- Although SOD2 expression correlates with TRPS1 expression, SOD2 is regulated by several transcription factors (such as FOXOs, NRF2, etc…).
- Despite authors claim that TRPS1 and miR-221 have antagonistic expression, levels of miR-221 in normal and degenerated IVD samples are not statistically different.
- In Figure 2, a small legend indicating the color code for the bar graphs would improve clarity.
- The method to analyze protein expression in Figure 2A is rather unusual. Why more traditional and better quantitative methods such as PCR or western blot were not used?
- SOX2 expression levels alone are not a good indication of cell stemness.
- Only chondrogenic genes were analyzed upon TRPS1 overexpression, but not specific genes more relevant to IVD biology. The role of TRPS1 in the maintenance of the IVD phenotype remains largely unexplored in this study.
Reviewer 2 Report
In this study, the authors investigated possible mechanisms underlying the inverse regulation between miR-221 and TRPS1, also exploring potential molecules that lie upstream and downstream of TRPS1 signaling, and that are involved on the ability of lumbar IVD cells to differentiate, maintain the antioxidant defense and stemness. Several opinions are advised:
1. The IVD is a complex joint consisting of NP, AF, and cartilaginous endplates. In this study, the authors used the mixture cells. Is it possible to investigate possible mechanisms in single type of cells?
2. Further studies are needed to investigate possible relationships of these molecules in disc degeneration animal model.
3. Further studies are needed to investigate possible relationships of these molecules in in different regions of the degenerated spine, ex: cervical and lumbar IVDs.
Reviewer 3 Report
In the work with the title “Reciprocal regulation of TRPS1 and miR-221 in intervertebral disc cells” the authors show, that decreased TRPS1 expression goes along with higher IVD degeneration. Concerning stress response, SOD2 protective for IVD, since it acts against oxidative stress by clearing mitochondrial reactive oxygen species, is lower expressed with increased IVD degeneration. Overexpression of TRPS1 in IVD primary cell cultures, used at P2 in a de-differentiated but not senescent state, shows increased level of chondrogenic marker expression, therefore a restorage of a chondrogenic like phenotype, which is lost in the de-differentiating process. Moreover, the authors could show, that TRPS1 and miR-221 expression are linked together, overexpression of TRPS1 in these primary cell cultures led to significantly reduced expression of miR-221. Furthermore, the authors of this manuscript could demonstrate that TRPS1 regulates miR-221 expression by binding to different promotor regions. The more IVD degeneration exists, the less binding sites for TRPS1 are accessible on the miR-221 promotor region. This group could also show, that vice versa miR-221 binds to the 3’UTR of TRPS1, regulating its expression. This work provides new promising targets in the therapy of IVD degeneration.
Overall the authors provided a very nice work, nevertheless I have some minor comments mainly concerning the language which in some parts is very good but appear in other parts with mistakes, therefore the authors should carefully edit the language in this manuscript.
Row ”46”: occur is without an s in that case and spina should be spine
Row “54”: citation 4 is in a different style than the rest
Row “56”: Consequently, detecting potential specific target therapeutic molecules should be Consequently, detecting potential specific therapeutic target molecules
Row “77”: What is meant with loop involving cells? This should be written more specific.
Row “81”: molecule instead of molecole
Row “231”: It is Vectastain ABC solution not Vecstain
Row “243-245”: Formatting needs revision
Row “286”: There is too much space before “All”
Figure1 A+B: The bar does not show a size, at least not in the version I got, if it is there it should be bigger.
Row “305”: The authors state that SOD2 plays an important role in cellular stress response, this should be written more precisely, like antiapoptotic and against oxidative stress, although it is discussed later, so it is easier for the reader to understand.
Row “324”: Before “In these” is too much space
Figure 2: Labelling of the white, black and grey bar is missing, which one is control, hTRPS1 and EMPTY
Figure 2 C: The authors show that on the one hand chondrogenic markers are higher expressed in TRPS1 transfected cells, but on the other hand that these cells also express higher level of SOX2 a stem cells marker, which is somehow confusing. Control staining of totally differentiated chondrocytes would be needed to compare expression intensity and in which range TRPS1 transfection leads to chondrogenic differentiation. Moreover, the authors state in the discussion that it was previously shown that there is a potential stem cell niche detectable in IVD cells, this should be mentioned in this part of the manuscript shortly, not only in the discussion otherwise it is misleading for the reader.
Row “340”: immunohystochemical should be immunohistochemical
Row “362”: potential TRPS1 binding sites, here the s was missing
Figure 3: The authors used only one patient for the respective degradation stage, to confirm the conclusions the authors made, it should be done at least in 3 patients of the same stage each.
Row “389”: Citation 16 is in a different style than the rest
Row “477-478”: Please change the sentence to: this extends our previous finding
Reviewer 4 Report
Comments:
I like the idea of the current study very much. Gene expression control happens at multiple levels and TRPS1-miR axis may be very important. However, the authors should describe more specifically about the significance of their study and why TRPS1-miR-221 axis is so critical? Is there any intervertebral disc phenotype in TRPS1 knockout mice or is there any spine intervertebral disc phenotype because of TRPS1 mutation in human? In method section 2.2, I am not sure about the Streptomycin concentration (100mg/mL or 100ug/mL??). In method section 2.4, please mention the species you cloned the 739 bp UTR region of TRPS1 mRNA, also with the positions of the nucleotides. Is this region containing the SEED seq for miR-221-3p or 5p please indicate their positions? (http://www.targetscan.org/cgi-bin/targetscan/vert_72/view_gene.cgi?rs=ENST00000395715.3&taxid=9606&members=&showcnc=0&shownc=0&showncf1=&showncf2=&subset=1#miR-221-3p/222-3p)? In method section 2.8, I am curious whether miR-221/222 is inter or intragenic? There are three species of non-coding RNA transcribed from opposite strand. Did authors have any idea what happens to miR-222 and lncRNA miR222HG expression in IVD cells? (https://useast.ensembl.org/Homo_sapiens/Gene/Summary?g=ENSG00000207870;r=X:45746157-45746266;t=ENST00000385135). Figure 1, Please replace Fig.1 (A & B) with higher quality. Figure 2, western blot needed for TRPS1 in panel A and a known target for miR-221 in Panel B. Figure 3, fold enrichment over IgG does not specify percent binding over input. IgG nonspecific binding may be varied depending on the regions. I want to see the % binding over input using =100*2^(adjusted input CT-IP CT). Do authors have data on promoter activity (reporter assay) using promoter region 3 or region 1 including WT and mutation in the TRPS1 binding site? I also want to see ChIP assay for the level of H3K4me3 or H3K27ac modifications in region 3 and H3K36me3 modification on region 1 to support miR-221 transcription repression and reduced transcription elongation. Figure 4D, I am confused about how anti-miR-221 degrading endogenous miR-221-3p? I thought anti-miR blocks the loading of the corresponding miR on to the RISC complex resulting inhibition. Do you know the mechanism of miR-221 degradation because of miR-221 antagomir treatment? The same binding site of TRPS1 mRNA is targeted by miR-222-3p and TRPS1 transcriptionally controls both of them, did you check level of miR-222-3p. Finally, TRPS1 expression and control of miR-221 transcriptional fate and miR-221 expression and TRPS1 post transcriptional fate are highly and tightly regulated by tissue specific fashion. Any idea, how this fine balance is maintained? How you study is going to decipher that tissue specific balance?
Reviewer 5 Report
A very well documented paper, containing the needed elements for the reviewer, as well as for an interested reader, to understand why, how and the outcomes of this study.
Round 2
Reviewer 4 Report
Comments against revision:
Revision: As shown in the Figure, using % input, the binding profile of the analyzed samples evaluated for the different regions of miR222/221 promoter, does not change.
Question: Still the maximum binding is 0.15%. . I am not sure that this percent binding is epigenetically significant. I need additional data to show that this regulation is significant. The authors can address in suitable cell line also. Show me TRPS1 binding results in histone modifications.
